# Advancements and Challenges in Ancient DNA Research: Bridging the Global North–South Divide

**DOI:** 10.3390/genes14020479

**Published:** 2023-02-14

**Authors:** Vasundhra Dalal, Nagarjuna Pasupuleti, Gyaneshwer Chaubey, Niraj Rai, Vasant Shinde

**Affiliations:** 1Centre for Cellular and Molecular Biology, Hyderabad 500007, Telangana, India; 2Cytogenetics Laboratory, Department of Zoology, Banaras Hindu University, Varanasi 221005, Uttar Pradesh, India; 3Ancient DNA Lab, Birbal Sahni Institute of Palaeosciences, Lucknow 226007, Uttar Pradesh, India

**Keywords:** ancient DNA, evolution, adaptation, migration, archaeogenetics

## Abstract

Ancient DNA (aDNA) research first began in 1984 and ever since has greatly expanded our understanding of evolution and migration. Today, aDNA analysis is used to solve various puzzles about the origin of mankind, migration patterns, and the spread of infectious diseases. The incredible findings ranging from identifying the new branches within the human family to studying the genomes of extinct flora and fauna have caught the world by surprise in recent times. However, a closer look at these published results points out a clear Global North and Global South divide. Therefore, through this research, we aim to emphasize encouraging better collaborative opportunities and technology transfer to support researchers in the Global South. Further, the present research also focuses on expanding the scope of the ongoing conversation in the field of aDNA by reporting relevant literature published around the world and discussing the advancements and challenges in the field.

## 1. Introduction

The primal evidence for understanding the past living forms and environment comes from instances of direct observation, the existence of homologies and fossils and certain biogeographical patterns. The first comprehensive theory of evolution, published in the middle of the 19th century by Charles Darwin and Alfred Russel Wallace, substantially offered a distinctive arena for academicians to delve into the past to understand the present [1]. Subsequently, until the 1980s, the majority of the research work generated was regarding evolutionary processes inferred through morphometric analysis. Eventually, the possibility of exploring the genetic extent of past events to provide direct historical evidence opened a novel line of research in 1984. 

The molecular Investigation of ancient DNA (aDNA) was initiated in 1984 when a team of researchers was able to isolate and sequence a short DNA fragment from the dried muscle of a museum specimen of the quagga, a South African equid (Equus quagga) that became extinct at the beginning of the twentieth century [2]. This initial aDNA study employed bacterial cloning to amplify short sequences retrieved from the skin of the animal. The very next year, another breakthrough research conducted by Pääbo et al. recovered a 3.4 kilobases long DNA fragment from the dried tissue of an Egyptian mummy that was 2430 years old [3]. These studies, nonetheless, in a way, were premature as the DNA from ancient samples can have a high degree of damage or have contaminated DNA along with it, which will make it difficult to isolate and clone the specific DNA sequences as desired. However, the advancements in molecular biology tools, primarily polymerase chain reaction, fueled aDNA investigations [4,5]. The PCR enabled the generation of an infinitely large number of copies from very few or even a single original DNA. Henceforth, several copies of the same DNA sequence from the same specimen could be amplified, allowing for the scientifically rigorous study of ancient DNA. 

Now, more than thirty decades later, scientists and researchers are probing to solve various puzzles regarding the origin of mankind, migration patterns, the advent of infectious diseases and their spread throughout ancient populations. aDNA analysis has emerged as a cutting-edge genetic tool altering our comprehensive understanding of the past, gaining impetus from all over the world. For instance, in recent breakthrough research, aDNA analysis of seven individuals revealed the origin of the plague strain that caused the Black Death in present-day Kyrgyzstan [6]. In fact, from identifying “lost” Indigenous populations to encountering new branches within the human family, including that of the Denisovans, close relatives of Neanderthals, has been aided by aDNA [7,8]. Many ancient animals, such as woolly mammoths and cave bears, have had their genomes sequenced, as have human tribes, ranging from Vikings to Paleo-Eskimos to Neanderthals [9,10,11,12,13].

Today, with the help of aDNA, the existing atlas of genetic variety is no longer confined to a glimpse of the diversity observed in modern-day populations throughout the globe. Rather, it is constantly updated with data sets that monitor changes in human, animal, plant, and even microbial populations’ genetic ancestries as they grow, implode, and adapt to new environmental factors [14,15,16]. Apart from biological and archaeological prospects, aDNA also holds the potential to establish powerful political and cultural connections with other nations [17]. More broadly, aDNA data have revolutionized our knowledge and curiosity, leading to the publication of enormous amounts of literature. The latest innovations in technology have further proved to be an exceptional tool for scientists and researchers in the race to fetch the “gold” (aDNA). In the present research, our aim is to expand the scope of ongoing conversations in the field of aDNA by reporting the relevant literature published around the world and elaborating on the advancements and challenges in the field.

## 2. Rise of aDNA Field

aDNA has developed from a niche area that has recently been transformed into a very lucrative and rising area of study and has truly advanced our understanding of the evolutionary history of many species. Ever since the emergence of aDNA analysis, bone and teeth have been the most researched substrates to retrieve an efficient amount of DNA from the specimen. The majority of paleontological hard tissues consist of relatively low levels of endogenous DNA [18]. As a result, there has been a great deal of research on techniques (including column-based extractions) to extract endogenous DNA from bone specimens and from other skeletal components, such as the petrous bone and dental cementum [19,20,21,22,23]. The frequently used next-generation sequencing or NGS in association with enriched capture-based techniques has, however, been responsible for resolving the primary constraints on aDNA research concerning time and expense [18,24].

Most commonly, the fields comprising aDNA analysis are also known as palaeogenomics or archaeogenomics. While the number of studies employing aDNA over the past few decades has substantially escalated, in this context, the present study notably managed to illustrate the trend of aDNA studies consisting of human and non-human samples (Figure 1 and Figure 2). 

It is evident from Figure 2 what aDNA research has yielded so far. The studies screened for the present research highlights the branching of aDNA into various fields, such as anthropology, archaeology, geology, botany, zoology, paleopathology, microbiology, molecular biology, and genetics and developing a symbiotic relationship. This amalgamation has outlined some broad areas where aDNA sequences have provided novel insights.

### 2.1. Species Phylogenies

The potential to link extinct and extant species via molecular phylogenies is one clear study path offered by ancient DNA analysis. Australian marsupial wolves [25], New Zealand moas [26], American ground sloths [27] and endemic Hawaiian geese [28] are prominent examples of animal species that are now extinct. Even several natural history museums have created standards for removing samples for molecular studies and have even set up molecular laboratories to work on their collections, acknowledging that they are genetic repositories [29]. This has helped researchers to obtain nuclear DNA sequences from various Pleistocene animals and from plants that are preserved in dry environments [30,31,32].

The evolutionary connections of various non-human primates, however, are still unknown. This is because those primates were primarily located in hot, humid climates with acidic soil, such as tropical rainforests, where DNA storage is poor [33]. Many of the recently extinct Malagasy lemurs’ evolutionary connections have also been investigated utilizing aDNA methods. As several of Madagascar’s now-extinct lemur species lived until as recently as 500 years ago, they can be considered evolutionary contemporaries of living species [34]. Nevertheless, the study of extinct lemurs has already provided results utilizing ancient mitogenomes that have cast doubt on existing morphologically based lemur phylogenies [35]. The research in this direction opens the door to a number of new avenues for learning more about the previous history of macaques in Europe and apes on mainland Asia throughout the Late Pleistocene, as well as their subsequent local extinction [33].

### 2.2. Migration and Phylogeography

The support of the “Recent African Origin” idea of modern human origins, first put out on the basis of fossil data, is one of the most important findings of mtDNA analysis [36]. Various research on mtDNA variation in populations have constantly discovered further evidence for this hypothesis, with the most common ancestor of human mtDNA located in Africa about 100,000–200,000 years ago [37,38]. Furthermore, precise mtDNA analysis of Neanderthal and their contemporaries, early modern humans from Europe, revealed quintessential results ruling out any significant genetic contributions from Neanderthals to early modern humans, but it does not preclude a lesser contribution [39,40]. A comprehensive knowledge of past human migrations shed light on significant events in human expansion—peopling of the New World, the colonization of the Pacific, and the initial migration to New Guinea and Australia [41,42,43,44].

Recent publications have also contributed to this specific sub-domain of aDNA studies by solving the long-standing debate on ‘Anglo-Saxon settlement’ [45]. The results published demonstrated that the mediaeval society in England must have been the result of mass migration across the North Sea. A study conducted in 2019 revealed surprising results, suggesting that during the Neolithic period, the spread of farming in Eurasia resulted in the formation of a gradient in southwestern Asia that stretched from the Neolithic to the Bronze Age [46]. This gradient was characterized by a mix of ancestry related to Anatolian farmers in the west and Iranian farmers in the east. These findings have important implications for our understanding of the history and evolution of human populations in this region [46].

Despite the rich history and cultural diversity of South Asia, the prehistoric genomic details of this region have remained largely unknown due to a lack of ancient DNA data. However, several studies have recently emerged to shed new light on the complex origins and genetic ancestry of modern South Asian populations, providing valuable insights into the migratory history, admixture, and genetic diversity of the region. Moreover, ancient DNA studies have provided strong evidence for a mixture between European and South Asian populations [47,48]. A study by Allentoft et al. (2015) analyzed the genomes of 101 ancient individuals from across Europe and found that there was significant gene flow between Europe and South Asia during the Bronze Age (c. 3000–1000 BCE). This gene flow is thought to have been driven by the expansion of the Steppe pastoralists, who migrated from the Pontic-Caspian Steppe (located in modern-day Ukraine and Russia) into Europe and South Asia [47]. In addition to the Steppe pastoralists, there is also evidence of gene flow between Europe and South Asia during the Indus Valley Civilization [47]. Moreover, a study analyzing genome-wide data from ancient individuals in eastern Iran, Turan, Bronze Age Kazakhstan, and South Asia has provided some insight into the complex origins of South Asians [49]. This study suggests that there was a southward migration from the Eurasian Steppe to South Asia around 2300–1500 BCE. Another study represented the evidence of admixture between Ancestral North Indians (ANI) and Ancestral South Indians (ASI), which is present in the modern Indian populations, including tribal groups from the Andaman Islands [50,51].

Further, adding to the curiosity to understand the Indus Valley Civilization, Shinde et al. (2019) reported the first genome-wide data from the specimen excavated at the Harappan site of Rakhigarhi, Haryana, India. Such findings have made significant contributions to our knowledge of the South Asian genetic history, the spread of agriculture throughout South Asia, and Indo-European languages [52,53]. The startling discovery of relics of queen Keetvan in India and numerous skeletons in the Roopkund lake was the start of pioneer studies conducted in India that strengthened the roots of the aDNA field in India [17,54]. Adding to the field, the latest research on the Ajnala massacre showcased the implication of aDNA in population migration and forensic science by employing isotope dating methods along with aDNA analysis to identify the origin of the specimens [55].

### 2.3. Archaic Hominis

The abundance of newly accessible ancient human genetic data has been helpful in shifting our picture of the human past from one of long-term population continuity and isolation to one in which movement and population mixing have played considerably major roles [16]. The most crucial contribution of aDNA is towards solving the complex puzzle—Neanderthal Man. It is now evident that the progenitors of all modern non-African populations intermingled with Neanderthals [56]. The recent genetic evidence suggests that all modern non-African populations possess approximately 2% Neanderthal ancestry [57,58,59]. Furthermore, through the analysis of linkage disequilibrium, researchers have determined that this event of hybridization occurred around 50–65 thousand years ago, providing a vital time frame for early human migration [60]. A study examining the intricate history of interaction between Neanderthals and modern humans has shown that the East Asian population has a higher concentration of Neanderthal sequences (approximately 20%) compared to the European population [58,61,62]. This difference underscores the impact of natural selection and the occurrence of additional hybridization events in the ancestors of present-day East Asians after the separation from the European population.

Apart from this, a small finger bone discovered in the Denisova Cave in Siberia led to a publication of ground-breaking research in 2010 [8]. This bone’s morphology was insufficient to determine if it came from a contemporary human, a Neanderthal, or something else. When the nuclear DNA of this bone was examined, however, it revealed a unique scenario: the group represented by this bone is a sister group of Neanderthals that separated from them after modern humans’ ancestors parted from Neanderthals [8]. This was further confirmed by analyzing a high-coverage genome obtained from the same fossil [63]. Denisovan have a unique hybridization history with contemporary people. It was also reported that Melanesians and aboriginal Australians have Denisovan heritage, as do other East Asians to a lesser extent [64]. Although Denisovans are only known from a solitary cave in Siberia, their mixing pattern implies that they formerly inhabited a larger area [64]. The examination of partial genomes recovered from two teeth discovered in the Denisovan Cave supports this result [65].

### 2.4. Paleopathology

aDNA from microorganisms, including pathogens and commensals, can give insights about peoples’ health, as well as shifts in diet and ecology of the diseases. Initially, the research on ancient microbes used PCR-based techniques to identify pathogens (for example, pathogens responsible for causing skeleton lesions characteristic of leprosy) and also to analyze the 1918 influenza virus. However, PCR-based techniques failed to distinguish ancient and modern contaminating microbial DNA, which was overcome by the NGS-based techniques [66].

The contribution of palaeopathology has unveiled crucial information about most of the diseases that have significantly affected the past population or are still persistent in the present society. For instance, aDNA analysis from samples with congenital hip dislocation in the mediaeval community of Notre Dame du Bourg in Digne Les Bains, France, revealed that the condition, which is more common in this group, was caused by a mechanical element rather than a genetic (hereditary) factor [67]. In India, the first instance of leprosy in ancient skeletal remains was found in a man from the second millennium [68]. Furthermore, the prevalence of TB in America prior to the arrival of Europeans has been confirmed by the finding of Mycobacterium tuberculosis in a 1000-year-old pre-Columbian mummy [69]. The very same bacteria were discovered in Egyptian skeletons dating from 2500 to 5000 years ago, in British mediaeval bones, and in Hungarian skeletons dating from 300 to 1300 years ago [70,71,72]. All this evidence has enabled researchers to study the development of Mycobacterium TB and its interactions with its hosts [73].

Moreover, when Yersinia pestis was detected in the teeth of 400-year-old plague victims [74] while Mycobacterium leprae (leprosy agent) was discovered on the bones of 300-year-old archaeological German and Hungarian sites [70], the analysis from the victims of both the Plague of Justinian and the black Death demonstrated that Y. pestis was the causal agent in both the dreadful pandemics [75,76]. Despite a plethora of research that has achieved widespread recognition in the field of archaeoparasitology, there have not been many notable reports from South Asian nations. However, in a short report published in 2018, an attempt to examine parasites on Indian archaeological specimens collected from the Mature Harrapan site and Chalcolithic sites was made, proposing the possibility of conducting paleoparasitological studies in India [77]. Such exploration in ancient pathogen genomic divergence and its evolutionary histories could help researchers to reconstruct and study the origins of present-day geographical distribution and diversity of clinical pathogen infections, as well as recognize potential pathogen evolutionary responses caused by environmental factors [78]

### 2.5. Adaptation

From several studies, it has been identified that the capacity to analyze how people have adapted to climatic change through time is a core promise of aDNA research. aDNA analysis has also highlighted the significance of Late Pleistocene climatic cycles, as well as representing how the Holocene and modern environments’ generally stable circumstances may be regarded rather abnormal when examining recent evolutionary events [79]. Early research revealed that some epigenetic alterations, for example, methylation of cytosine, may be examined in ancient material [80]. This now leads us to explore the changes in epigenetic markers in large numbers of samples from populations undergoing substantial environmental changes over time [81]. 

An interesting take on adaptation further leads us to discuss a study published in 2019, where ancient remnants found in a lake situated in the Himalayas not only questions the reasons behind their death but also their presence in the area at such an altitude [54]. At 5029 km above sea level, where oxygen levels are low, it becomes nearly impossible to breathe unless they have a sort of adaptability to such an extreme environmental condition [54]. Henceforth, there are numerous questions for archaeologists, anthropologists and geneticists to address, which would pave the way for interesting opportunities not only to examine extreme human adaptations but also the evolution of microbiomes as species adapt to different climates and diets.

## 3. Technological Advancements

Currently, researchers are processing hundreds of specimens in what has been compared to a “factory-like method” of extracting ancient DNA [82]. This explosion of research in aDNA could only be possible due to improvements in extraction methods and sequencing technologies. 

### 3.1. Revolutionized Extraction Methodologies

Since the majority of recent ancient genetic research has focused on bone and teeth, extraction techniques have been adjusted to enhance the recovery of endogenous DNA [83,84]. Because the knowledge about DNA preservation that we have is meagre, there is an exigency for the improvement in DNA extraction techniques from the tissues. Generally, when there are enough materials (ancient specimens) for comprehensive analysis, researchers started to inquire about the potency of various extraction methodologies for decayed plants [85], dental calculus from the archaeological specimens [86], parchments [87] and coprolites [88]. With regard to exotic archaeological artefacts, the analysis is dependent on contemporary experimentation or artificially damaged materials [89]. It is imperative that the challenges in authenticating aDNA from unusual materials (ancient specimens) put forward a captivating scenario for “unified protocols” to extract a variety of genetic information [88,89]. 

Further, to extract crucial information in the form of DNA, destructive methodologies have been adopted by researchers around the world. Because of this, museums and archives are urging to safeguard the archaeological specimens for future analyses by opting for non-destructive sampling [90,91]. Considering this, research has created unique sampling methods that do not need intrusive procedures for bones, teeth, insects, shells, parchment, as well as shells [92,93,94,95]. We are optimistic that with more advancements in extraction methodologies, a whole new arena of information will be revealed, even from minute samples. 

### 3.2. Evolution in Sequencing Technology

With the invention of PCR, researchers were profoundly able to target and replicate specific DNA sequences. Before 2010, researchers substantially used PCR along with Sanger sequencing to retrieve ancient genomic sequences [89]. However, the limitations that were observed in the former application were overcome by the development of the next-generation sequencing (NGS). NGS made it possible for the researchers to analyze the evolutionary changes in the plethora of species through time and sequence the entire mitochondrial genome along with the reconstruction of nuclear genomic regions [96,97]. This pioneering technology has further redefined aDNA research through its technical applications, such as DNA hybridization capture [18,98]. Hybridization capture has proved to be fruitful specifically in rejuvenating research on ancient pathogen DNA as an active area of study [99]. In addition, for the non-permafrost samples where even shotgun sequencing drastically failed, DNA hybridization made it possible to recover the earliest aDNA, which was from a ~ 400,000-year-old archaic hominin [20]. 

An encouraging approach to enable aDNA analysis attainable to a substantially greater number of laboratories is to utilize the “1240k capture” method on ancient samples. This “1240k capture” panel is advantageous, as it increases efficacy by focusing on specific regions of the human genome that will be examined while providing access to genome-wide data from ancient samples with minimal amounts of human DNA [100]. Another added advantage of hybridization capture is that it can be used in both solid and liquid phases. Both procedures incorporate washing away non-specific DNA fragments, eluting target DNA fragments from the probes, and sequencing the enriched libraries using NGS systems. Interestingly, capture methods can be utilized to produce accurate nuclear sequence data even when endogenous DNA concentration is less than 0.03% since they are appropriate for short DNA fragments that are frequently maintained in ancient materials [101]. Despite being primarily used to analyze modern DNA, capture technologies have been successfully employed to retrieve the entire genome of a mediaeval Yersinia pestis sample as well as to query hundreds of SNPs in the Neanderthal genome [24,75].

### 3.3. Advancements in Data Analyzing Tools

Tools for data analysis have significantly transformed the study of aDNA. Depending upon the nature of the research objective, that is, if it is focusing on extracting information on population history modeling, microbiological profiling, or paleo-environmental reconstruction, various pipelines and tools could be adopted [14]. Assessing contemporary populations for variations seen in aDNA is one of the fundamental techniques for locating ancient haplotypes. This straightforward method has been successfully used to pinpoint contemporary populations in Europe where aDNA samples revealed mtDNA mutations, providing an estimate of populations/regions that contain such ancient genetic fingerprints [102,103]. Astonishing results from the ancient sample came forward with the single-nucleotide polymorphism (SNPs) analysis that revealed primordial characteristics such as skin, hair and eye pigmentation along with identification of sex by determining the ratio assigned to the Y- and X- chromosomes [104,105]. 

As more and more ancient human remains are retrieved with continuously advancing technologies, researchers have gained a better understanding of the genetic diversity within populations by utilizing admixture-based techniques on contemporary and historical samples [106]. When combined with anthropological information and historical records, tools and procedures originally created for population research of modern people can be used to retrace migration routes, places of origin, and local and global ancestries of extinct populations. The exploratory statistical methods have further elevated the aDNA field by condensing the genetic variations, representing the genetic affinities [107,108]. Adding to this, rarely does endogenous DNA (1%) appear in ancient DNA (aDNA) libraries; environmental DNA often takes up the majority of sequencing capacity. The aDNA obtained from the biological sample consists of a short length with frequent dominance of environmental microbial DNA contamination; however, the processed reads are now being utilized not only in reference to the targeted species but also to possible microbial pathogens that the individuals may have been exposed to throughout his or her life [14]. Moreover, aDNA extracts typically exhibit a full metagenomic diversity of ambient microorganisms that mostly colonize the subfossil material after death [10,109]. In this context, numerous bioinformatic pipelines are designed to map and estimate the source of microorganisms, such as soil, gut, and oral microbiota [14]. Further developments in bioinformatic tools could have the propensity to raise the resolution of evolutionary events by considerably aiding whole-genome sequencing. 

## 4. Dominance of Global North in aDNA Research

Going further in the research, it is equally important to understand the status of aDNA studies around the world. We utilized the web of science database to accumulate articles on aDNA studies around the globe and plotted them on the world map. It became apparent from the observation (Figure 3 and Figure 4) that there are clearly significant differences between the Global North (GN) and Global South (GS) (the terms used to classify nations according to their socioeconomic and political characteristics) in terms of research outputs [110]. The underlying premise that could be drawn from such results is that aDNA is a specific area of research where asymmetries among infrastructure, funding support, and training for research could create this division—GS (lower-income countries) and GN (higher-income countries) [111]. The researchers from the GN (with the exceptions of Australia and New Zealand from GS) frequently travel to the GS to gather data and materials for processing, analyzing, and publishing findings with little to no local collaboration leading towards developing a whole new approach referred to as “helicopter research” or “parachute research” [112,113]. Sometimes, this unfair collection of samples (without collaboration or proper acknowledgements) leads to a breach in ethical concerns and ultimately shows no remorse towards local beliefs.

The lack of aDNA research output from the lower-income countries could possibly be due to operating costs associated with establishing aDNA laboratories, which also, down the line, hinders the sustenance of the laboratory. Additionally, for many developing countries, the aDNA does not necessarily take precedence over financial and health crises. Moreover, the plight of the researchers from tropical countries is further exacerbated, as in these regions the DNA preservation of the ancient biological remains is not amenable, because of which very little data on ancient DNA have been generated [64,114,115]. In this regard, the most well-established laboratory in the world has the capability to utilize their powers to support researchers from Global South, as they are hardly encouraged to determine the research objective [109]. As the majority of laboratories around the world are still struggling to have access to the latest technologies, the transfer or exchange of technologies could further prove to be a boon [116].

## 5. Challenges to Overcome

Although astonishing results appeared in Figure 2. depicting a gradual increase in the exploration of non-human and human aDNA studies, respectively, the subject is rife with challenges and pitfalls [14].

### 5.1. Availability of aDNA from the Ancient Biological Samples

The enzymatic repair systems protect DNA molecules from chemical damages that often occur in live cells, preserving the genome’s integrity [117]. These cellular restoration processes cease to work once the host is dead. The genome is consequently subjected to the full effects of several causes that endanger its stability. These causes include intracellular nucleases, which are no longer locked away inside the cell and can thus access and break down DNA, as well as microbes that spread through decomposing tissues. All recoverable DNA may be lost as a consequence of these forces working together. However, under unfavorable environmental circumstances, such as when tissues freeze or quickly dry up after death, these processes are slowed down before all DNA that is endogenous to the organism is completely destroyed. Even though it is not difficult to retrieve ancient DNA from warmer places, the quality of the genetic material is typically much worse [118,119]. Adding to this, the duration of DNA survival in tissue is then limited by additional damaging mechanisms, especially hydrolytic and oxidative activities and with the increasing age of ancient remains [120,121].

However, if preventive measures taken by archaeologists at the site can further help in limiting contamination. It is necessary to accurately document the archaeological site and findings, as it could further help geneticists and anthropologists to understand if there were chances of contamination. When dealing with biological specimens, archaeologists should not use the same equipment. Along with this, while working on the burial site, only a single site should be targeted at once to avoid any environmental contamination or exposure to sunlight.

### 5.2. Contamination and Authenticity of aDNA

The environmental settings around the ancient remains are abruptly altered during the excavation. The chances of experiencing greater thermal stress are the highest because of the frequent exposure to sunlight for long periods, specifically in tropical regions [118]. Moreover, without protective gloves, handling the ancient biological remains will leave the excavator’s DNA on the surface of the specimen. While excavating ancient human remains, it is necessary to wear personal protective equipment (PPE), as ancient human remains are particularly vulnerable to modern human DNA contamination. It is because of these preventive measures that, at the Harappan site of Rakhigarhi, India (a tropical region where preservation conditions are not favorable), archaeologists and scientists were able to extract aDNA from the specimen (Figure 5) [52]. aDNA can also be hampered despite having a good preservation environment and excavation techniques. For example, research conducted in 2008 observed a unique instance of contamination in the specimen (cattle); as the bones were excavated under pristine circumstances, several samples were tainted with goat DNA [122]. Even after the successful excavation of the specimen, it is crucial to provide optimal storage facilities (reducing the chances for DNA damage) such as electric cool boxes, plastic bags and cool, dry and dark conditions.

As aDNA is highly sensitive towards contamination, it requires a laboratory setting with positive air pressure, UV surface irradiation, stringent cleaning techniques (bleach treatment of surface) and filtered air systems to minimize contamination [14,123]. However, maintaining such facilities costs a hefty amount of money, which most institutes/countries could not afford and hence resulted in a limited number of aDNA laboratories around the globe. Further, the variety of statistical analyses that may be performed on genome-wide sequence data is essentially infinite, which can get obstructed by contaminations, damage by post-mortem DNA, and read alignment [14]. Unsurprisingly, the debunking of a number of claims regarding the extraction of valid DNA from different tissues has led to a great deal of criticism and cynicism toward the subject of aDNA. To reduce this widespread contamination and damage in highly deteriorated samples, studies then might have been confined to reads exhibiting post-mortem damage or restricted to transversion, which dramatically impacts the availability of the data [10,124,125]. However, concrete modeling accompanied by quantitative analysis of contamination and damage can help in validating the findings [126]. 

### 5.3. Ethical Concerns

Oftentimes paleo-genomics research poses ethical and cultural issues that should be carefully considered before, during, and following the research. Even the destruction of relics or defiling of holy places has brought up some serious concerns from ethnic communities. For instance, the case of Kennewick Man led to a huge feud between the Indigenous communities and the researchers [127]. Further, researchers flocking to aDNA-based research have raised concerns about addressing the ethical issues. Moreover, the genetic findings at times might result in the repression of Indigenous communities either by communal and personal convictions or by perpetuating chauvinist concepts [110]. Therefore, researchers have taken the initiative to publish papers accompanied by remarks outlining the efforts with which the research team have addressed the ethical issues [128,129]. 

Dealing with human remains becomes a sophisticated matter, as human remains have ingrained values, holding cultural beliefs of many communities in addition to being significant for solving scientific questions. Such concerns have led to producing certain guidelines to form an alliance between researchers and Indigenous communities, which includes addressing cultural and ethical implications, developing strategies to communicate and handle the data, engaging the Indigenous section of the society and formulating strategies for long-term responsibilities [130,131]. For example, in the USA, the Native American Graves Protection and Repatriation Act (NAGPRA), requires consultation with Indigenous communities to transfer ancient human remains [131]. Hence, to enrich the research process, it is advised prioritizing community engagement (Indigenous scholars and stakeholders) [130].

## 6. Mapping out the Future Direction

The capacity to identify and recreate the genetic archaeological record will undoubtedly enrich the human narrative and contribute to the transdisciplinary scientific endeavor of comprehending our shared past in the future. In recent decades, aDNA research has expanded and matured in reaction to breakthroughs in the social sciences and humanities and has progressed in lockstep with technical and computational advancements in the biological sciences. The realization that a vast and unseen molecular history exists in the archaeological evidence has profoundly influenced archaeologists, geneticists, and anthropologists, promising a slew of fresh and unexpected discoveries in the future. 

From our perspective, more than ever, it is now crucial to address the concern regarding collaborations between researchers, especially from Global North and South. The sharing of technology and equipment is the need of the hour for the researchers in the Global South. Supporting local researchers through collaborative initiatives and providing international grants for research can further boost strong links among the research communities. Apart from that, institutes or nations that have funds to set up aDNA laboratories, and researchers working in highly advanced aDNA facilities should come forward with efficient protocols, contributing to escalating research outputs in the field of aDNA [132]. Another aspect that should also be brought into consideration is organizing workshops, seminars and conferences that are more inclusive. As the cost of being a participant is very high, reducing it or providing financial assistance to researchers enables them to be a part of the global collective discourse. Furthermore, focusing on Global South, educational institutes, apart from mainstream courses, should also emphasize interdisciplinary courses such as palaeogenetic/archaeogenetic, biological anthropology and population genetics, further facilitating a greater influx of researchers in the ever-growing field of aDNA.

Recently, it has also been noticed that a few labs from Global South have been exceptionally producing ground-breaking research. For instance, India has marked its position in producing quintessential aDNA discoveries [17,52,54,55,133,134]. Being able to extract aDNA from the biological specimen in such climatic conditions signifies the extraordinary precaution taken in the field and in the laboratories. Apart from this, the genetic diversity present within India has caught the attention of the world, as it stands as an epitome for studying various diseases, admixtures, and migration [51,52,54,135,136,137]. 

The study of ancient DNA has the potential to provide answers to long-standing questions about the origins of certain populations. Such as when did West Eurasian groups first arrive in South Asia and mix with most Indian people? Moreover, the presence of South Asian ancestry in modern European populations has important implications for the study of disease susceptibility and population-specific traits. Understanding the genetic contributions of South Asian ancestry to modern European populations can help researchers identify genetic risk factors for certain diseases and develop targeted therapies. Certainly, by utilizing advanced techniques for analyzing ancient DNA and working in conjunction with archaeologists and anthropologists, it is possible to delve into the past and reconstruct the population history around the world. Further, this approach may allow us to gain a deeper understanding of the events and influences that have shaped the genetic makeup of the modern Indian population after the collapse of the Harappan civilization.

The field of aDNA has reached a significant milestone after the historic recognition of Svante Paabo’s paleogenetic research on Neanderthals by Nobel Prize in 2022. Fortunately, in recent years, scholars have increasingly learned to collaborate and have developed several initiatives to establish and strengthen the critical interdisciplinary dialogue by encouraging extensive symbiotic association between archaeologists and geneticists/anthropologists to elevate the domain knowledge forming the foundation for future reflective archaeogenetics.

## Figures and Tables

**Figure 1 genes-14-00479-f001:**
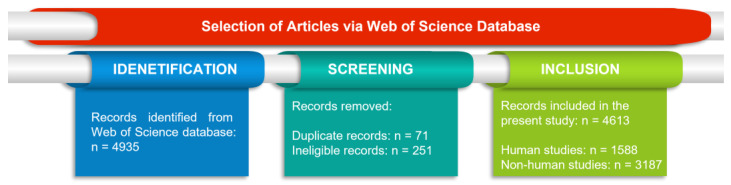
The studies extracted from Web of Science (date: 1 August 2022) containing the terms (palaeogenom* OR paleogenom* OR archaeogenom* OR “archaic DNA” OR archeogenom* OR “Ancient DNA” OR (archeology* NEAR/1 DNA) OR (archaealogy* NEAR/1 DNA) OR “extinct DNA” OR (historic* NEAR/1 DNA) OR (paleo* NEAR/1 DNA) OR (palaeo* NEAR/1 DNA). Further, the studies were manually examined to verify the authenticity of the derived data. The screening criteria include checking the relevance by title, abstract and keywords.

**Figure 2 genes-14-00479-f002:**
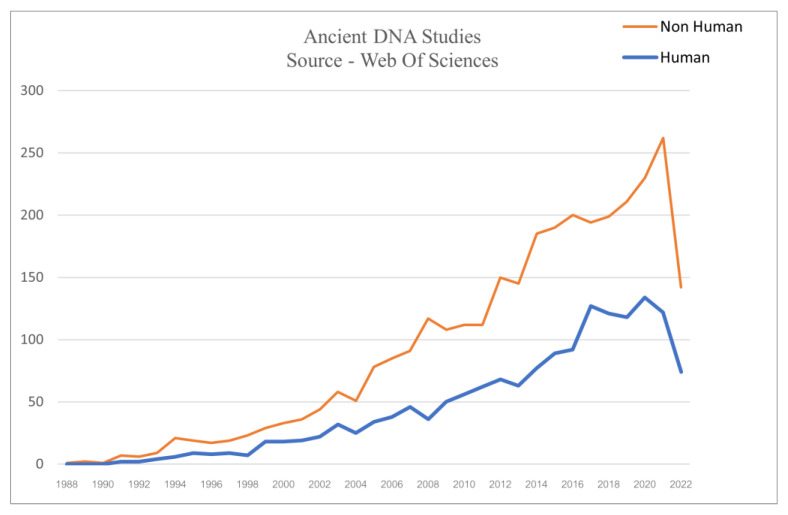
The total number of aDNA research articles (encompassing Sediments, Archaeological Artefacts and Ecofacts, Calcified and Mineralized substrates and Biological and Cultural archives) consists of 3187 non-human studies and 1588 human studies.

**Figure 3 genes-14-00479-f003:**
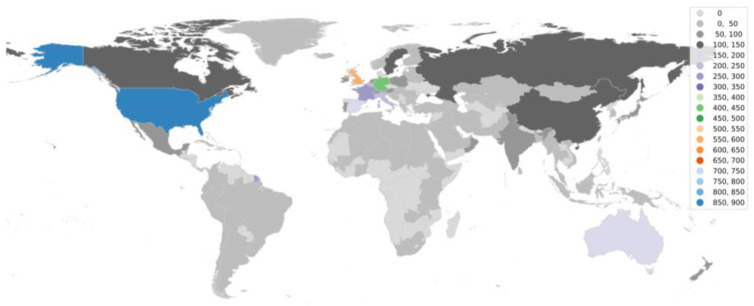
World map showing aDNA human-based research publications around the globe.

**Figure 4 genes-14-00479-f004:**
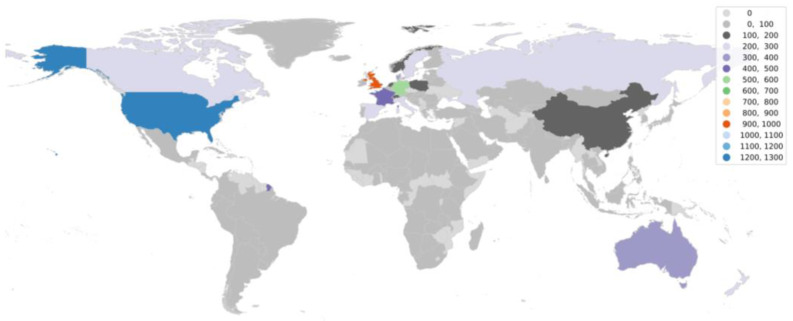
World map showing aDNA non-human-based research publications around the globe.

**Figure 5 genes-14-00479-f005:**
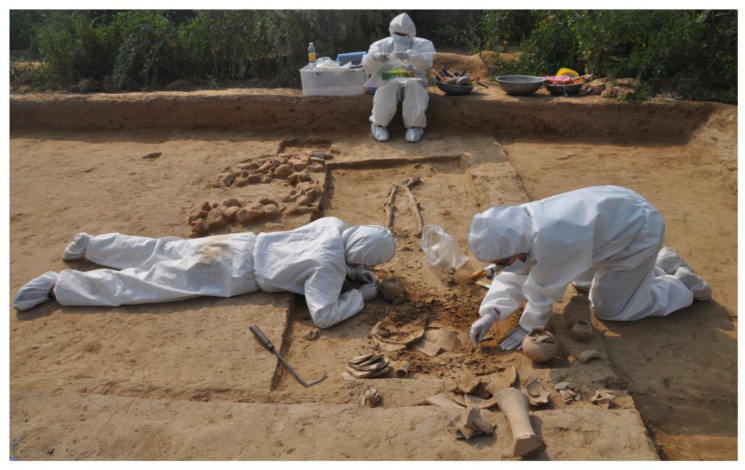
Contamination-avoiding measures taken by the excavators at the Harappan site of Rakhigarhi.

## Data Availability

Not applicable.

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
