# Peer review of "Advancements and Challenges in Ancient DNA Research: Bridging the Global North–South Divide"

_genes, 2023, doi:10.3390/genes14020479_

Round 1

Reviewer 1 Report

The study by Dalal et al. provides an overview of the field of ancient DNA, including historical developments, challenges, and recent applications. It is very well-written and informative, and I enjoyed reading it. I particularly appreciate the highlighting of the disparity between research in the Global North and the Global South. I only have a few minor comments:

1) The sentence starting at line 142 with "A greater..." seems grammatically incorrect.

2) Line 176: Referring to the Indus Valley Civilization as "the world's finest major civilization" seems very odd -- this is very subjective. Such a statement should not be included in a scientific paper.

3) In chapter 2.3. (Archaic Hominins) I miss a major finding from aDNA -- that modern non-African individuals have about 2% of Neanderthal admixture. The way it is currently written, it sounds like aDNA is rejecting any admixture.

4) Lines 259-261: It seems questionable to me to speculate that epigenetic modifications have anything to do with environmental adaptation and dynamics of megafauna. There is currently no evidence supporting this, and the evidence from extant mammals seems very weak.

5) There are some minor grammatical errors throughout the manuscript -- I recommend additional careful proofreading.

Reviewer 2 Report

This article does a great job at highlighting the advancements and challenges in the field of ancient DNA research, particularly in regards to bridging the Global North and South divide. However, it focuses almost exclusively on human ancient DNA and does not discuss plant and other organisms ancient DNA studies. A better title for this article could be: Advancements and Challenges in Ancient DNA Research: Bridging the Global North-South Divide. Overall, the quality of the article is very high and as far as I can understand from reading it (I am not an expert in human ancient DNA but only plant ancient DNA), it seems comprehensive and well written. I suggest to change the title.
